# Robust Multi-agent Counterfactual Prediction

**Alexander Peysakhovich**[*]
Facebook AI Research

**Christian Kroer**[*]
Facebook Core Data Science

**Adam Lerer**[*]
Facebook AI Research

## Abstract

We consider the problem of using logged data to make predictions about what would happen if we changed the 'rules of the game' in a multi-agent system. This task is difficult because in many cases we observe actions individuals take but not their private information or their full reward functions. In addition, agents are strategic, so when the rules change, they will also change their actions. Existing methods (e.g. structural estimation, inverse reinforcement learning) assume that agents' behavior comes from optimizing some utility or that the system is in equilibrium. They make counterfactual predictions by using observed actions to learn the underlying utility function (a.k.a. type) and then solving for the equilibrium of the counterfactual environment. This approach imposes heavy assumptions such as the rationality of the agents being observed and a correct model of the environment and agents' utility functions. We propose a method for analyzing the sensitivity of counterfactual conclusions to violations of these assumptions, which we call robust multi-agent counterfactual prediction (RMAC). We provide a first-order method for computing RMAC bounds. We apply RMAC to classic environments in market design: auctions, school choice, and social choice.

## 1 Introduction

Constructing rules that lead optimizing agents to good collective outcomes is the goal of the field of mechanism design (Roth and Peranson, 1999; Roth et al., 2005; Abdulkadiroğlu et al., 2005; Klemperer, 2002; Roth, 2002; Porter et al., 2003). Good mechanism design is particularly important for businesses which make their livelihoods as platforms (e.g. internet ad auctions, ride sharing, dating sites). A key challenge for designers in practice is to observe an existing set of rules at work and make a counterfactual statement about how outcomes would change if the rules changed (Bottou et al., 2013; Athey, 2015).

The multi-agent counterfactual question is difficult for two reasons. First, participants are strategic. An agent's optimal action can change due to changes in the rules, and often, can change when other agents change what they are doing. Second, agents have private information that is not known to the designer so even knowledge of the rules, and ability to compute optimal actions, is insufficient to estimate counterfactual outcomes. The analysis of online ad auctions provides a good example: if we observe data from a series of first-price sealed bid auctions and wanted to predict what would happen to revenue if we changed the auction format to second price with a reserve we would need to account for how agent behavior would change in response to these new incentives.

A common class of approaches to this question assume that observed actions are coming from a multi-agent system where all agents are optimizing some latent reward functions. In other words, that the system is in some form of Nash equilibrium. Further, they assume that once changes are made, the system will again equilibrate. Given these two assumptions, counterfactual prediction becomes a question of how equilibria change as the mechanism changes. Such assumptions are typical in the

---

[*]Equal contribution, author order has been randomized.

field of inverse reinforcement learning (Ng et al., 2000) and in structural estimation in economics (Berry et al., 1995; Athey and Nekipelov, 2010).

A downside of this approach is that it requires strong assumptions that are not always completely true in practice. For example, this process requires assuming that agents are optimizing their utility given the behavior of others so that an analyst can infer underlying 'taste' parameters from agent actions. It is well known, however, that human decisions do not always obey the axioms of utility maximization (Camerer et al., 2011) and that both mistakes and biases can persist even when there is ample opportunity for learning (Erev and Roth, 1998; Fudenberg and Peysakhovich, 2016).

The main contribution of this work is a method that computes a robust interval of counterfactual estimates under relaxations of the assumptions of rationality and correct specification of the model.

Our first contribution is to show that the counterfactual estimation problem corresponds to identifying equilibria in a game which we call a revelation game, and that the set of $\epsilon$-equilibria correspond to counterfactual predictions when assumptions are relaxed. We consider particular $\epsilon$-equilibria of the revelation game - the 'worst' and 'best' elements of the $\epsilon$-equilibrium set with respect to some evaluation function (e.g. revenue). These equilibria form the upper and lower bounds for our robust multi-agent counterfactual prediction (RMAC). [2] We show that computing the RMAC bounds exactly is a difficult problem as it is NP-hard even for 2-player Bayesian games.

As our second contribution, we propose a first-order method which we refer to as revelation game fictitious play (RFP) to compute the RMAC bounds and discuss its convergence properties.

Our third contribution is to apply RFP to generate RMAC in three domains of interest for mechanism designers: auctions, matching, and (in the Appendix) social choice. In each of them we find that some counterfactual predictions are much more robust than others. We also demonstrate that RMAC can be applied even when standard assumptions about point identification do not hold (e.g. when there are multiple equilibria or when the data is consistent with multiple type distributions) to compute optimistic and pessimistic counterfactual predictions.

## 1.1   Related Work

Our work is closely related to the notion of partial identification (Manski, 2003). The main idea behind partial identification is that many statistical models are only able to recover a set of parameters consistent with the data, not a single point estimate. The PI literature focuses on models where this 'identified set' can be extracted easily. The revelation game is strongly related in that the equilibrium relaxation we employ makes the counterfactual predictions a set rather than a point. We focus on finding this set's worst (in terms of some evaluation function) and best elements.

Existing work in the field of market design has used econometric techniques to estimate counterfactuals in specific applications (Athey and Nekipelov, 2010; Chawla et al., 2017; Agarwal, 2015). These approaches are, like ours, designed with the goal of answering counterfactual questions. However, while they allow for measures of statistical uncertainty they do not allow analysts to check for robustness of conclusions to violations of assumptions. Haile and Tamer (2003) consider using 'incomplete' models of auctions to provide some form of robustness but, like much of the literature on the econometrics of auctions (and unlike RMAC), requires hand-deriving estimators specifically tailored to the auction at hand.

Since the pioneering work of Myerson (1981) there is a large subfield of game theory dedicated to designing mechanisms that optimize some quantity (e.g. seller revenue). Myerson-style results often require the auctioneer to know the distribution of types (valuations) in the population. These strong assumptions are relaxed in robust mechanism design (Bergemann and Morris, 2005), automated mechanism design (Conitzer and Sandholm, 2002), and recent work in using deep learning methods to approximate optimal mechanisms (Dütting et al., 2017; Feng et al., 2018). Optimal mechanism design is related to, but different from, the RMAC problem as it typically assumes access to at least some direct information about the distribution of types, whereas the RMAC problem is to robustly

infer the underlying types from observed actions. However, these problems are related and combining insights from these literatures with RMAC is an interesting direction for future work.

There is recent interest in relaxing equilibrium assumptions in structural models. Nekipelov et al. (2015) consider replacing equilibrium assumptions with the assumption that individuals are no-regret learners. This, again, gives a set valued solution concept which can be worked out explicitly for the special case of auctions. Given the prominence of no-regret learning in algorithmic game theory a natural extension of the work in this paper is to consider expanding RMAC to learning as a solution concept.

## 2 Bayesian Games

We consider the standard one-shot Bayesian game setup. There are $N$ players which each have a type $\theta_i \in \Theta$ drawn from an unknown distribution $\mathcal{F}$. This type is assumed to represent their preferences and private information. For example, in the case of auctions this type describes the valuations of each player for each object.

**Definition 1.** *A game $\mathcal{G}$ has a set of actions for each player $\mathcal{A}_i$ with generic element $a_i$. After each player chooses their action, the players receive utilities given by $u_i^{\mathcal{G}}(a_1, \ldots, a_N, \theta_i)$.*

We focus on systems that come to a stable state, in particular we assume that they form a *Bayesian Nash equilibrium*. We denote a strategy $\sigma_i$ for player $i$ in game $\mathcal{G}$ as a mapping which takes as input $\theta_i$ and outputs an action $a_i$. As standard for a vector $x$ of variables, one for each player, we let $x_i$ be the variable for player $i$ and $x_{-i}$ be the vector for everyone other than $i$.

**Definition 2.** *An Bayesian Nash equilibrium (BNE) is a strategy profile $\sigma^*$ such that for each player $i$, all possible types $\theta_i$ for that player which have positive probability under $\mathcal{F}$, and any other strategy $\sigma_i'$ we have*

$$\mathbb{E}_{\mathcal{F}}\big[u_i^{\mathcal{G}}(\sigma_i^*(\theta_i), \sigma_{-i}^*(\theta_{-i}), \theta_i)\big] \geq \mathbb{E}_{\mathcal{F}}\big[u_i^{\mathcal{G}}(\sigma_i'(\theta_i), \sigma_{-i}^*(\theta_{-i}), \theta_i)\big].$$

The Bayesian Nash equilibrium (BNE) assumption can be motivated by, for example, assuming that repeated play (with rematching) have led learning agents to converge to such a state (Fudenberg and Levine, 1998; Dekel et al., 2004; Hartline et al., 2015). Importantly, BNE states that players' actions are optimal given the distribution of partners they could play, not necessarily that they are optimal at each realization of the game with types fixed.

For the purposes of lightening notation from here on we will deal with games where every player's action set is the same $\mathcal{A}_i = \mathcal{A}$ and every players' type is drawn iid from $\mathcal{F}$.

## 3 The Revelation Game as a Counterfactual Estimator

Given the formal setup above, we now turn to answering our main question:

**Question 1.** *Suppose we have a dataset $\mathcal{D}$ of actions played in $\mathcal{G}$. What can we say about what would happen if we changed the underlying game to $\mathcal{G}'$?*

Formally, when we say that we change the game to $\mathcal{G}'$ we mean that the action set changes to $\mathcal{A}'$ and the utility functions change to $u_i^{\mathcal{G}'}(a_1, \ldots, a_N, \theta_i)$. $\mathcal{G}'$ remains a Bayesian game so the definitions and notation above continue to apply.

As a concrete example: in the case of online advertising auctions, $\mathcal{D}$ will contain a series of auctions with bids taken by different participants. We may wish to ask, what would happen if we changed the auction format? It is important to note here that $\mathcal{D}$ only contains actions played in the game and not types (which are never observed by the analyst).

We now discuss a set of assumptions typically made either implicitly or explicitly in existing literature. We will refer to these as the *standard assumptions*.

**Assumption 1** (Equilibrium). *Data is drawn from a BNE of $\mathcal{G}$ and play in $\mathcal{G}'$ will form a BNE.*

**Assumption 2** (Identification). *For any possible distribution of types $\mathcal{F}$ and associated BNE $\sigma^*$ there does not exist another distribution of types $\mathcal{F}'$ and BNE $\sigma'^*$ that induces the same distribution of actions.*

**Assumption 3** (Uniqueness in $\mathcal{G}'$). *Given $\mathcal{F}$ there is a unique BNE in $\mathcal{G}'$.*

If the standard assumptions are satisfied then the counterfactual question can be answered as follows. By Assumption 1 each action $d_i$ is optimal against the distribution of actions implied by $\mathcal{D}$. If $\mathcal{D}$ is large enough then it approximates the true distribution implied by $\sigma$ and $\mathcal{F}$. By Assumption 2, there is a unique $\sigma$ and $\mathcal{F}$ that give rise to this distribution. Therefore, $\mathcal{F}$ can be estimated using $\mathcal{D}$, and the equilibrium in $\mathcal{G}'$, which is unique by Assumption 3, can be estimated using standard methods.

We now show this procedure is equivalent to solving for the Nash equilibrium in a modified game which we refer to as a *revelation game*.[3] We do not consider that agents will actually play this game, rather we will show that this proxy game is a useful abstraction for doing robust counterfactual inference.

The revelation game has $m$ players, one for each element of $\mathcal{D}$. We refer to these as data-players to avoid confusion with the players in $\mathcal{G}$ and $\mathcal{G}'$. Each data-player knows that the analyst has a random variable $\mathcal{D}$ of actions from the equilibrium of $\mathcal{G}$. $\mathcal{D}$ includes the data-player's own true equilibrium action but the other actions are ex-ante unknown. Each data-player has a true type $\theta_j$ which is unknown to the analyst, the types of the other data-players $-j$ are unknown to $j$ but it is commonly known that they are drawn from the distribution $\mathcal{F}$.

Each data-player $j$ makes a decision: they report a type $\hat{\theta}_j$ and an action for the counterfactual game $\hat{a}_j$. They are paid as follows: first, let the $\mathcal{D}_{-j}$ denote the random variable which denotes the actions of the other data-players the analyst will observe. Now we define the $\mathcal{G}$-Regret of data-player $j$ as

$$\text{Regret}_j^{\mathcal{G}}(\hat{\theta}_j, \mathcal{D}_{-j}) = \max_{a_j} \mathbb{E}\big[u_j^{\mathcal{G}}(a_j, \hat{\theta}_j, \mathcal{D}_{-j})\big] - \mathbb{E}\big[u_j^{\mathcal{G}}(d_j, \hat{\theta}_j, \mathcal{D}_{-j})\big].$$

We define the $\mathcal{G}'-$Regret of data-player $j$ as

$$\text{Regret}_j^{\mathcal{G}'}(\hat{a}_j, \hat{\theta}_j, \hat{a}_{-j}) = \max_{a_j} \mathbb{E}\big[u_j^{\mathcal{G}'}(a_j, \hat{\theta}_j, \hat{a}_{-j})\big] - \mathbb{E}\big[u_j^{\mathcal{G}'}(\hat{a}_j, \hat{\theta}_j, \hat{a}_{-j})\big].$$

The revelation game is a Bayesian game where each data-player $j$ tries to minimize a loss given by the max of the two above regrets:

$$\mathcal{L}_j^{rev}(\hat{\theta}_j, \hat{a}_j, \hat{a}_{-j}, \mathcal{D}) = \max\{\text{Regret}_j^{\mathcal{G}}(d_j, \hat{\theta}_j, \mathcal{D}), \text{Regret}_j^{\mathcal{G}'}(\hat{a}_j, \hat{\theta}_j, \hat{a}_{-j})\}.$$

Given these definitions, we can show the following property:

**Theorem 1.** *If assumptions 1-3 are satisfied then the revelation game has a unique BNE where each agent reveals their true type and counterfactual action.*

We leave the proof of the theorem to the Appendix. This property means that if we can solve for the equilibrium of the revelation game, then we have our counterfactual prediction. With this result in hand, we now discuss how to modify the revelation game to make our counterfactual predictions robust.

## 4   Robust Multi-agent Counterfactual Inference

In reality, assumptions 1-3 above are rarely satisfied exactly and we would like to see how robust conclusions are to violations of these assumptions. In addition, all modeling makes the important assumption

**Assumption 4** (Specification). *$\mathcal{G}$ and $\mathcal{G}'$ include the correct specifications of individuals' reward functions.*

which, like the others, is rarely completely true in practice.

To relax all of these assumptions we will consider the concept of $\epsilon$-BNE. $\epsilon$-BNE requires that, given the behavior of individuals $-i$, the decision of each individual $i$ yields at most $\epsilon$ regret relative to the optimal strategy. Formally this replaces the inequality in definition 2 by $\mathbb{E}_{\mathcal{F}}\big[u_i^{\mathcal{G}}(\sigma_i^*(\theta_i), \sigma_{-i}^*(\theta_{-i}), \theta_i)\big] \geq \mathbb{E}_{\mathcal{F}}\big[u_i^{\mathcal{G}}(\sigma_i'(\theta_i), \sigma_{-i}^*(\theta_{-i}), \theta_i)\big] - \epsilon.$

Allowing for $\epsilon$-BNE in the revelation game means that we are also allowing for $\epsilon$-BNE in $\mathcal{G}$ and $\mathcal{G}'$ since the revelation game loss is defined as the maximum of the two regrets. The introduction of $\epsilon$-BNE is how we relax assumptions 1-4. In the Appendix we give a longer and more formal treatment of the relationship between $\epsilon$-BNE and the assumptions. Informally, notice that $\epsilon$-equilibria can arise because agents are imperfect optimizers[4] (but are able to learn to avoid actions that cause huge negative regret) or because the utility functions in $\mathcal{G}$ or $\mathcal{G}'$ are slightly incorrect (and individuals reach an equilibrium corresponding to some other reward function).

However, like many instances of partial identification Manski (2003) $\epsilon$-BNE is a set valued solution concept. Rather than enumerate the whole set, we will consider particular boundary equilibria:

We assume the existence of an evaluation function $V(\theta, a)$ which gives us a scalar evaluation of the counterfactual outcome that the analyst cares about. We overload notation and let $V(\sigma) = \mathbb{E}_{(\theta,a)\sim\sigma} V(\theta, a)$ be the expected value of $V$ given a mixed strategy $\sigma$. Common examples of valuation function used in the mechanism design literature include revenue, efficiency, fairness, envy, stability, strategy-proofness, or some combination of them (Roth and Sotomayor, 1992; Guruswami et al., 2005; Budish, 2011; Caragiannis et al., 2016).

We will consider the maximal and minimal elements of the $\epsilon$-BNE set with respect to $V$. Formally:

**Definition 3.** *The $\epsilon$-pessimistic counterfactual prediction of $V$ is*

$$\inf_\sigma V(\sigma) \ s.t. \ \sigma \ is \ an \ \epsilon\text{-BNE in the revelation game.}$$

*The $\epsilon$-optimistic prediction replaces the inf with sup. The $\epsilon$-RMAC bounds are the values of $V$ attained at the pessimistic and optimistic predictions.*

The figure to the right summarizes the idea behind RMAC. The structural assumptions imply a one-to-one mapping between observed distributions and underlying types followed by a one-to-one mapping between underlying types and counterfactual behavior. Assuming only $\epsilon$-equilibrium makes both of these mappings one-to-many and RMAC bounds select the most optimistic and pessimistic counterfactual distributions consistent with these mappings.

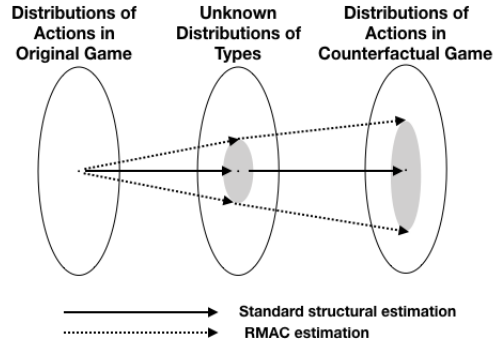

## 5 Computing Equilibria of the Revelation Game

In practice, we can replace the random variable $\mathcal{D}$ of the revelation game with their sample analogue, the observed data. From here forward $\mathcal{D}$ will refer to the sample data. Unfortunately, we can derive a negative complexity result for computing $\epsilon$-RMAC bounds exactly:

**Theorem 2.** *It is NP-hard to compute the robust counterfactual estimate even if each data-point $j$ has only a single feasible type, and there are only two data points. It is also NP-hard even if there is no objective function, a finite number of feasible types, and $\mathcal{G}'$ has only two players.*

The proof is provided in the Appendix. In the Appendix we also provide a mathematical program with equilibrium constraints for the case of pure-strategy $\epsilon$-BNE, and a mixed integer program for the special case of two-player games.

Given that computing RMAC bounds for the general case requires solving a mathematical program with equilibrium constraints, we do not expect it to scale beyond small instances. Therefore, we propose to adapt the fictitious play algorithm Brown (1951) to compute the optimistic and pessimistic equilibria of the revelation game. We refer to this as *Revelation Game Fictitious Play (RFP)*.

RFP works as follows. For notation, let $\hat{\theta}_i^t$ be the estimated type for data point $i$ at iteration $t$ and $\hat{a}_i^t$ be the estimated counterfactual action at iteration $t$. As with standard fictitious play, at each time step

each $i$ takes an action in the revelation game (i.e. reports a type-action pair). They observe the choices of others and update their $t+1$ choice $(\hat{\theta}_i^{t+1}, \hat{a}_i^{t+1})$ to be the one that minimizes (or maximizes) $V$ out of the set of $\epsilon$ best responses to the current history of play (when $\epsilon = 0$ RFP simply chooses the best response to the current history, breaking ties randomly). The pseudocode is shown in Algorithm 1.

---

**Algorithm 1** Revelation Fictitious Play

---

**Input:** $\epsilon, \mathcal{D}, V, \mathcal{G}, \mathcal{G}'$, if pessimistic then $\alpha = -1$, if optimistic then $\alpha = 1$
Randomly initialize $\hat{\theta}_i^0, \hat{a}_i^0$
**for** $t = 0, \ldots$ while not converged **do**
    Let $\bar{a}_{-i}^t$ be the historical distribution of $\hat{a}_{-i}^{t'}$ for $t' \in \{0, \ldots, t\}$
    Let $\bar{\sigma}_{-i}^t$ be the (mixed) strategy profile implied by the historical distribution of $(\hat{\theta}_{-i}^{t'}, \hat{a}_{-i}^{t'})$
    Let the set of low-regret revelation game actions be

$$\hat{\mathcal{C}}_i^t = \{(\hat{\theta}_i, \hat{a}_i) \in \Omega \times \mathcal{A} \mid \mathcal{L}_i^{rev}(\hat{\theta}_i, \hat{a}_i, \bar{a}_{-i}^t, \mathcal{D}) \leq \epsilon\}$$

    Breaking ties randomly, update guesses for each datapoint

$$(\hat{\theta}_i^{t+1}, \hat{a}_i^{t+1}) = \operatorname{argmax}_{\hat{\theta}_i, \hat{a}_i \in \hat{\mathcal{C}}_i^t} \left[ \alpha V(\hat{\theta}_i, \hat{a}_i, \bar{\sigma}_{-i}^t) \right].$$

---

It is well-known that fictitious play converges in 2-player zero-sum and potential games, while it may cycle in general. Nonetheless, a well-known result states that *if* fictitious play converges, then it converges to a Nash equilibrium (Fudenberg and Levine, 1998).

We now show an analogous result for RFP: if pessimistic (optimistic) RFP converges then it converges to an $\epsilon$-BNE and locally minimizes (maximizes) $V$ in the sense that no unilateral deviation by a single data-player $j$ in the revelation game that are *strictly* $\epsilon$-best responses leads to a smaller (bigger) $V$.

We denote by $\bar{\sigma}^t$ the *mixed strategy* implied by the history of play. As with standard fictitious play we consider convergence of $\bar{\sigma}^t$:

**Definition 4.** *RFP converges to a mixed strategy $\sigma^*$ if $\lim_{t \to \infty} \bar{\sigma}^t = \sigma^*$.*

We use the following notion of local optimality (analogously defined for optimistic V):

**Definition 5.** *A mixed $\epsilon$-BNE $\sigma^*$ of the revelation game is* locally V-optimal *if*

$$V(\sigma^*) \leq V(\theta_j, a_j, \sigma_{-j}^*)$$

*for any data-player $j$ and unilateral deviation $(\theta_j, a_j)$ where[5]*

$$\mathbb{E}_{(\theta_{-j}, a_{-j}) \sim \sigma_{-j}^*}[\mathcal{L}_j^{rev}(\theta_j, a_j, a_{-j}, \mathcal{D})] < \epsilon.$$

**Theorem 3.** *If RFP converges to $\sigma^*$ then $\sigma^*$ is a locally V-optimal $\epsilon$-BNE of the revelation game.*

We relegate the proof to the Appendix. The argument is an extension of standard fictitious play results to the revelation game.

An important question is whether RFP can be guaranteed to converge in particular classes of Bayesian games. We leave the theoretical study of RFP (or other learning algorithms in the revelation game) to future work and focus the rest of the paper on empirical evaluation.

# 6 Experiments

We now turn to constructing RMAC bounds for classic problems in market design. In the next two sections we discuss auctions and school choice. In the Appendix we consider two other experiments: 1) an auction setting where point identification is impossible and 2) social choice.

## 6.1 RMAC in Auctions

We first evaluate RMAC by studying counterfactual revenue in auctions. We consider a first-price 2-player auction $\mathcal{G}$ with types drawn from $[0, 1]$ uniformly and bids in the interval $[0, 1]$ discretized at intervals of $.01$. As our counterfactual games we consider a 2-player second-price auction with varying reserves[6] in the interval $[0, 1]$ and $N$ player first-price auctions.

We use counterfactual expected revenue as our evaluation function. We set the domain of possible types to also be equal to $[0, 1]$. We generate data by first sampling 1000 independent types and their actions from the closed form first-price equilibrium ($bid = .5\theta$), using these actions as $\mathcal{D}$. We then use $\mathcal{D}$ to compute $\epsilon$-RMAC predictions for several levels of $\epsilon$. Figure 1 shows our results with (small) error bars being shown as standard deviations of the statistic over replicates.

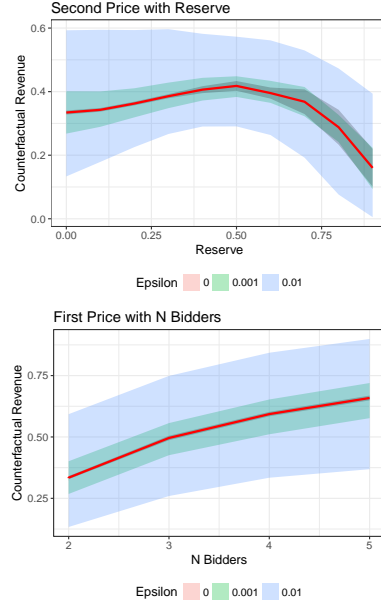

In Figure 1, we see that in auctions, even slight changes to $\epsilon$ can lead to larger changes in revenue. In particular, if we consider that the average expected utility accrued to the winner in the 2 player auction is $.25$, an $\epsilon$ of $.01$ corresponds only to a $4\%$ misoptimization/misspecification. However, this small $\epsilon$ still gives quite wide revenue bounds.

To see the logic behind this lack of robustness, consider the pessimistic estimate, in which the data is drawn from an $\epsilon$-equilibrium where individuals are overbidding in the original game and underbidding in the counterfactual game. Assuming a uniform bid distribution for others, an individual's regret for (unilaterally) shading their bid by $\Delta$ is $\epsilon = \Delta^2/2$. This will decrease expected revenue decrease by $\Delta$. Therefore, we expect a worst-case $\epsilon$-equilibrium in the counterfactual game to decrease revenue by $\sqrt{2\epsilon}$. In addition, there will be a similar decrease in revenue from the shift in types inferred from the original game.

Figure 1: RMAC revenue predictions using data drawn from the equilibrium of a first price 2 player auction for various counterfactual auction formats. The RMAC robustness bounds, even with small $\epsilon$ are much larger than the standard error bounds (grey ribbon around RMAC 0 line) estimated from multiple replicates.

In an additional experiment in the Appendix we further show that the robustness of counterfactual estimates for changing auction reserve price are assymetric. Specifically, counterfactual estimates for increasing the reserve are robust, while estimates for decreasing reserve are not.

## 6.2 RMAC in School Choice

We move to another commonly studied domain: school choice. Here the problem is to assign items (schools) to agents (students). Agents have preferences over schools, report them, and the output of the mechanism is an assignment.

We look at two real world school choice mechanisms. The first is the Boston mechanism (Abdulkadiroğlu et al., 2005). In Boston each student reports their rank order list and the mechanism tries to maximize the number of first choice assignments that it can. Once it has done this, it tries to maximize the number of second-choice assignments, and so on. The second mechanism uses the random serial dictatorship (RSD) mechanism (Abdulkadiroğlu and Sönmez, 1998). Here students are each given a random number and sorted, the first in line gets to choose their favorite school, the second chooses their favorite among what's left and so on.

The main tradeoff in practical school choice comes from balancing the total social welfare achieved by the mechanism and their truthfulness. RSD (and other algorithms like student-proposing deferred

acceptance) have a dominant strategy for each agent to report their true type. This means that participants in real world implementations of such mechanisms do not need to spend cognitive effort on guessing what others might do or searching for information - they can simply tell the truth and go on with their day. On the other hand, equilibria of the Boston mechanism can be more efficient in terms of allocating schools to students but in equilibrium need not be truthful (Mennle and Seuken, 2015; Abdulkadiroğlu et al., 2011).

We consider a problem mechanism with 3 students and 3 schools $(A, B, C)$. For both mechanisms the action space is a permutation over $A, B, C$.

We consider a hypothesis space of types that are permutations of utility vector $(5, 4, 0)$ - that is, individuals receive utility $5$ if they get their first choice, $4$ for the second and $0$ for the third. We are going to consider the case where all individuals have identical preferences of $A > B > C$. We will take these types, construct a dataset of equilibrium behavior under each mechanism, and ask what would happen if we switched to the other mechanism.

We examine two evaluation functions $V$: overall social welfare of the allocation and truthfulness of the strategies (i.e. whether types report their true values). We plot the estimated change in welfare and truthfulness from moving from one mechanism to another. This is an exercise that a market designer might perform in order to justify a change of mechanism.

Note that in the case of 'Boston to RSD' at $\epsilon = 0$ the standard structural assumptions are not satisfied, as multiple type distributions are consistent with the observed actions. Given our utility space, even though everyone has the same preferences, same types may choose different actions (i.e. play a mixed strategy), since it is better to be assured of getting $B$ than take a lottery between $A$, $B$ and $C$. So, some proportion of individuals will report $(B, A, C)$ However, such an action profile is also consistent with an equilibrium of truthful types with different preferences. Since the types are not identified from the observed actions, structural estimation using maximum likelihood has multiple optima with different values of $V$. However, RMAC with small $\epsilon$ will produce an interval

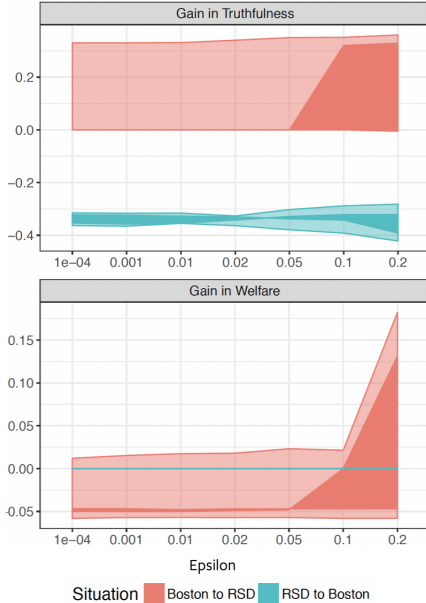

Figure 2: RMAC intervals for the change in social welfare and change in truthfulness from changing school choice mechanisms. Dark and light curves are for $10^{\text{th}}$ and $90^{\text{th}}$ percentile of estimated intervals over replicates with different sampled $\mathcal{D}$. The presence of multiple type distributions consistent with a given action distribution in Boston means that even for small $\epsilon$ RMAC bounds can be quite wide for Boston to RSD.

that covers both possible type distributions. As shown in Figure 2, switching from Boston to (truthful) RSD may increase truthfulness by 26% (if the types are indeed all $A > B > C$) or 0% (if the types matched the actions in Boston). Going from Boston to RSD also tends to lead to welfare decreases, although not always (e.g. if players have identical preferences, all mechanisms provide the same welfare). The counterfactual question in the other direction, RSD to Boston, has far tighter RMAC bounds because the types are well-specified by the truthful RSD mechanism.

# 7    Conclusion

Multi-agent counterfactual prediction is an important question both in theory and practice. We have introduced RMAC as a way of testing the robustness of counterfactual predictions with respect to violations of the standard assumptions of specification, equilibrium, point identification are not met.

Our method applies a version of fictitious play but it is well known that modifications to this algorithms can lead to large changes in real world performance (Conitzer and Sandholm, 2007; Syrgkanis et al., 2015; Kroer et al., 2015). In addition, more complex environments would require multi-agent learning algorithms that can handle function approximation such as those based on deep learning (Heinrich and Silver, 2016; Dütting et al., 2017; Lowe et al., 2017; Feng et al., 2018; Brown et al., 2018).

## Footnotes

[2]The RMAC bounds are different from standard uncertainty bounds (e.g. the standard error of a maximum likelihood estimator). Statistical uncertainty bounds (i.e. standard errors) reflect variance introduced by access to only finite data but still assume the underlying model is completely correct. On the other hand, our robustness bounds are intended to measure error that can come from the analyst using a model that is precisely incorrect but approximately true.

[3]We are indebted to Jason Hartline who pointed out in an earlier versions of this work that our optimization problem can be thought of as equilibrium finding and thus make exposition much simpler.

[4]Here the $\epsilon$ term is readily interpretable. For example, if our underlying game is an auction where bids are in US dollars then $\epsilon$ measures how many dollars an individual is giving up by playing their action instead of the best response.

[5]Note the strict inequality: the reason is that there may be deviations which have strictly greater than $\epsilon$ regret for all $t$, but their regret converges to $\epsilon$ from above, and so they enter the set at the limit.

[6]A reserve price $r$ in an auction is a price floor, individuals cannot win the auction if they bid below the reserve. In addition, in the case of second-price auctions, the price paid by the winner is the max of $r$ (as long as $r$ is less than the bid) and the second-highest bid.

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
