[Supplementary Material · RLMD_NeurIPS_Appendix.pdf]

# Appendix to Robust Multi-agent Counterfactual Prediction

**Alexander Peysakhovich**[*]
Facebook AI Research

**Christian Kroer**[*]
Facebook Core Data Science

**Adam Lerer**[*]
Facebook AI Research

## 1 The Relationship Between Structural Assumptions and $\epsilon$-BNE

Here we formalize the connection between RMAC bounds and $\epsilon$-BNE. In particular, we show that if individuals actually have some other utility functions than the ones we have explicitly modeled (this case also covers the situation where our modeled utility functions are correct but individuals do not perfectly optimize) then any equilibrium with respect to the real functions is an $\epsilon$-BNE of the revelation game with the modeled utility functions.

**Theorem 1.** *Let $(\mathcal{G}, \mathcal{G}')$ be the real game/counterfactual game, let $(\mathcal{G}_m, \mathcal{G}'_m)$ be misspecified versions of these two games with same type/action spaces but*

$$||u_m - u||_\infty \leq \frac{\epsilon}{2}$$

*and*

$$||u'_m - u'||_\infty \leq \frac{\epsilon}{2}.$$

*Let $\mathcal{D}$ be some data. If $r^* = (\hat{a}, \hat{\theta})$ is a BNE of the real revelation game corresponding to $(\mathcal{G}, \mathcal{G}', \mathcal{D})$ then $r^*$ is an $\epsilon$-BNE of the misspecified revelation game corresponding to $(\mathcal{G}_m, \mathcal{G}'_m, \mathcal{D})$*

*Proof.* We begin by considering $\mathcal{G}$ and $\mathcal{G}_m$. If $(\hat{a}, \hat{\theta})$ is an equilibrium of the revelation game then for any $j$ the

$$\text{Regret}_j^{\mathcal{G}}(\hat{\theta}_j, \mathcal{D}_{-j}) = 0.$$

We will show this implies that for any $j$ we have

$$\text{Regret}_j^{\mathcal{G}_m}(\hat{\theta}_j, \mathcal{D}_{-j}) \leq \epsilon.$$

To see this, pick some $j$ and let $m^*(\theta_j)$ be the optimal action in the misspecified game $\mathcal{G}_m$ for $\theta_j$. Since $(\hat{a}, \hat{\theta})$ is an equilibrium in the original revelation game we have that

$$u_j^{\mathcal{G}}(m^*(\theta_j), \hat{\theta}_j, \mathcal{D}_{-j}) \leq u_j^{\mathcal{G}}(d_j, \hat{\theta}_j, \mathcal{D}_{-j}).$$

By the sup-norm in the misspecification we can replace the left hand side by

$$u^{\mathcal{G}^m}(m^*(\theta_j), \hat{\theta}_j, \mathcal{D}_{-j}) - \frac{\epsilon}{2} \leq u^{\mathcal{G}}(d_j, \hat{\theta}_j, \mathcal{D}_{-j}).$$

We can also replace the right hand side by

$$u^{\mathcal{G}^m}(m^*(\theta_j), \hat{\theta}_j, \mathcal{D}_{-j}) - \frac{\epsilon}{2} \leq u_j^{\mathcal{G}_m}(d_j, \hat{\theta}_j, \mathcal{D}_{-j}) + \frac{\epsilon}{2}.$$

Subtracting gives the desired inequality:

$$u_j^{\mathcal{G}^m}(m^*(\theta_j), \hat{\theta}_j, \mathcal{D}_{-j}) - \epsilon \leq u_j^{\mathcal{G}_m}(d_j, \hat{\theta}_j, \mathcal{D}_{-j}).$$

---

[*]Equal contribution, author order has been randomized.

Thus regret in $\mathcal{G}_m$ is less than $\epsilon$.

We can repeat the same exercise with $\mathcal{G}'$. Let $m^*(\theta_j)$ be the best response to $\hat{a}_{-j}$ for type $\theta_j$ in the misspecified game $\mathcal{G}'_m$. By assumption of equilibrium of $(\hat{a}, \hat{\theta})$ with respect to the original games we have that

$$u_j^{\mathcal{G}'}(m^*(\theta_j), \hat{\theta}_j, \hat{a}_{-j}) \leq u_j^{\mathcal{G}'}(\hat{a}_j, \hat{\theta}_j, \hat{a}_{-j}).$$

Again we can use the sup norm bound to get

$$u^{\mathcal{G}'_m}(m^*(\theta_j), \hat{\theta}_j, \hat{a}_{-j}) - \frac{\epsilon}{2} \leq u^{\mathcal{G}'_m}(\hat{a}_j, \hat{\theta}_j, \hat{a}_{-j}) + \frac{\epsilon}{2}.$$

Rearranging gives that the regret in $\mathcal{G}'_m$ is less than $\epsilon$.

Since regret in both the misspecified games is less than $\epsilon$ we have shown that $(\hat{a}, \hat{\theta})$ is an $\epsilon$ equilibrium for the revelation game with misspecification. QED.

$\square$

## 2   A Mathematical Program for the General Revelation Game

We now present a mathematical program for solving the revelation game exactly for small instances. Throughout we will treat $V$ as a black box, assumed to be representable in the same class as the mathematical program it is stated within. Similarly we will assume that the Regret functions are representable within the given class. If these assumptions are not true then the problem will of course be harder than the stated class of mathematical programs.

Throughout the section we will abuse notation slightly in the name of readability and say that $u(a, a_{-j}, \theta_j) = \mathbb{E}_{\tilde{a} \sim a_{-j}}[u^{\mathcal{G}'}(a, \tilde{a}, \theta_j)]$, i.e. the expected utility of action $a_j$ given the distribution over actions taken by other players in $\mathcal{G}'$ given the action assignment of the data-players.

First, we give a mathematical program for solving the general case of the revelation game. Here we let $a$ and $\theta$ be vectors of action and type choices, since this formulation is guaranteed to have a pure-strategy BNE:

$$
\begin{aligned}
\min_{\theta, a} \quad & V(\theta, a) \\
\text{s.t.} \quad & \max_{a \in \mathcal{A}} u(a, a_{-j}, \theta_j) - u(a_j, a_{-j}, \theta_j) \leq \epsilon \quad \forall j \in \mathcal{D} \\
& \text{Regret}_j^{\mathcal{G}}(\theta_j) \leq \epsilon \quad \forall j \in \mathcal{D} \\
& \theta_j \in \Theta, a_j \in \mathcal{A} \quad \forall j \in \mathcal{D}, a \in \mathcal{A}, \theta \in \Theta
\end{aligned}
\tag{1}
$$

The first constraint in (1) is an equilibrium constraint over $\mathcal{G}'$, and therefore the general problem is a *mathematical program with equilibrium constraints* (MPEC). Thus the general program is quite hard. If we make the assumptions that $\mathcal{A}$ and $\Theta$ are nonempty convex sets, and each $u(\cdot, a_{-j}|\theta_j)$ is a concave function in the choice of action $a_j$ then we can formulate the problem as a variational inequality problem:

$$
\begin{aligned}
\min_{\theta, a} \quad & V(\theta, a) \\
\text{s.t.} \quad & \langle a' - a, F(a, \theta) \rangle \leq \epsilon \\
& \text{Regret}_j^{\mathcal{G}}(\theta_j) \leq \epsilon \quad \forall j \in \mathcal{D} \\
& \theta_j \in \Theta, a_j \in \mathcal{A} \quad \forall j \in \mathcal{D}, a \in \mathcal{A}, \theta \in \Theta
\end{aligned}
\tag{2}
$$

where $F(a, \theta_j)_j = \nabla_{a_j} u(a, \theta_j)$ is the gradient operator of $u$ for the given choice of $\theta$.

## 3   A Mixed Integer Program for Two Player $\mathcal{G}'$

Next, we give a mixed integer program (MIP) for the special case where $\mathcal{G}'$ has only two players, but where we may have an arbitrary finite number of data points. Furthermore, for this MIP we assume that $\Theta$ is discrete and finite, as well as $\mathcal{A}$ is finite.

The program has a Boolean variable $T_j^\theta$ for each pair of data point $j$ and type $\theta$, indicating whether data point $j$ takes on type $\theta$. For each data point $j$ and action $a$ we have $\sigma_j(a) \in [0, 1]$ indicating the

probability that $j$ puts on $a$ (we could make $\sigma_j(a)$ Boolean instead in order to compute a pure-strategy solution, but pure-strategy solutions are not guaranteed to exist when types are discrete).

We also have the following $\epsilon$-BNE-enforcing variables: $v_\theta$ represents the utility achieved by type $\theta$ in $\mathcal{G}'(T)$ under the computed solution, the slack variable $s_{\theta,a}$ denotes the *inoptimality* of $a$ when taken by type $\theta$, and $\delta_{\theta,a}$ is an indicator variable denoting whether $a$ is played by any data-player taking type $\theta$. The idea of the MIP is to ensure $s_{\theta,a} \leq \epsilon$, i.e. that inoptimality is bounded by $\epsilon$, whenever any data-player chooses type $\theta$ and puts nonzero probability on $a$.

$$
\begin{aligned}
\min_{T_j^\theta, \sigma(a), v_\theta, s_{\theta,a}, \delta_{\theta,a}} \quad & V(T, \sigma(a)) \\
\text{s.t.} \quad s_{\theta,a} - M\delta_{\theta,a} &\leq \epsilon && \forall a \in \mathcal{A}, \theta \in \Theta \\
\sigma_j(a) + T_j^\theta - \delta(\theta, a) &\leq 1 && \forall j \in N, a \in \mathcal{A}, \theta \in \Theta \\
\sum_{j',a'} \sigma_{j'}(a') u(a, a', \theta) + s_{\theta,a} &= v_\theta && \forall j \in N, a \in \mathcal{A}, \theta \in \Theta \quad (3) \\
\sum_{a \in \mathcal{A}} \sigma_j(a) &= 1 && \forall j \in N \\
\sum_{\theta \in \Theta} T_j^\theta &= 1 && \forall j \in N \\
T_j^\theta, \delta_{\theta,a} \in \{0,1\}, \sigma_j(a), s_{\theta,a} &\geq 0 && \forall j \in N, a \in \mathcal{A}, \theta \in \Theta
\end{aligned}
$$

Note that since $\Theta$ is finite we can preprocess it and remove all $\theta$ such that $\mathcal{R}_j^G(\theta) > \epsilon$, and thus we do not need to enforce this constraint on $T_j^\theta$ in the MIP.

## 4   Proofs of Theorems

*Revelation Game Equilibria Correspond to Assumptions.* The proof relies heavily on the fact that the revelation game's utility function is defined with respect to *regret* **not** the *original utility function*. Suppose that a data-player has true type $\theta_j$ but reports $\theta_j'$. In revelation-game BNE this $\theta_j'$ must have zero regret. But this violates the identification assumption, since we could then construct a new distribution $\mathcal{F}'$ where we reassign type $\theta_j$ to $\theta_j'$ but keep the same distribution over actions in $\mathcal{G}$ as part of a BNE. Thus the reported distribution over types must be $\mathcal{F}$ in revelation-game BNE. Now we can use the uniqueness assumption to infer that each data-player reports their true type, as well as their action in the unique BNE of $\mathcal{G}'$ given distribution $\mathcal{F}$. If they report any other action they must have nonzero regret, or they would violate the uniqueness assumption. $\square$

**Definition 1.** *An $\epsilon$-Bayesian Nash equilibrium is a strategy profile $\sigma^*$ such that for each player $i$, all possible types $\theta_i$ for that player which have positive probability under $\mathcal{F}$, and any other strategy $\sigma_i'$ we have*

$$
\mathbb{E}_\mathcal{F}\big[u_i^\mathcal{G}(\sigma_i^*(\theta_i), \sigma_{-i}^*(\theta_{-i}), \theta_i)\big] \geq \mathbb{E}_\mathcal{F}\big[u_i^\mathcal{G}(\sigma_i'(\theta_i), \sigma_{-i}^*(\theta_{-i}), \theta_i)\big] - \epsilon.
$$

*Solving Revelation Games Exactly is NP-Hard.* The first statement is by reduction from max-social-welfare Nash equilibrium in some game $G^{SW}$, which is NP-hard (Conitzer and Sandholm, 2008). We set $\epsilon = 0$, and $V(\theta, a)$ equal to the negative social welfare in $G^{SW}$ of actions $a$. For each agent in the NE problem we instantiate a data point $d_i$ and create the game $G$ such that each $i$ can only take on the type corresponding to their payoffs in $G^{SW}$ (this is easily done by making every other type have non-zero regret in $G$). Now we set $G' = G^{SW}$. A solution to the RMAC problem now corresponds to a social-welfare maximizing Nash equilibrium of $G^{SW}$.

The second statement is by reduction from the problem of checking whether a pure-strategy BNE exists, which is NP-complete (Conitzer and Sandholm, 2008). Consider a symmetric game $G^{pure}$ that we wish to find a pure-strategy BNE for. We let $G' = G^{pure}$. For each type $\theta$ of $G^{pure}$ we instantiate a data point such that only $\theta$ is a feasible type. Now the distribution over types in $G'$ equals that of $G^{pure}$, and so the equilibria are in correspondence. $\square$

*If RFP Converges then the Distribution is a BNE.* First we show that the limit $\sigma^*$ is an $\epsilon$-BNE. Let $(\bar{\theta}, \bar{a})$ denote a sequence of play in question. Denote by $\bar{\sigma}_j^t$ the strategy of player $j$ implied by the history $(\bar{\theta}, \bar{a})$ up to time $t$. We will use the notation $\mathcal{U}_j^{rev}(\hat{\theta}_j, \hat{a}_j, \hat{a}_{-j}, \mathcal{D}) = -\mathcal{L}_j^{rev}(\hat{\theta}_j, \hat{a}_j, \hat{a}_{-j}, \mathcal{D})$.

Suppose $\sigma^*$ is not an $\epsilon-$BNE. Then there exists data player $j$ and revelation game actions $(\theta_j, a_j)$ and $(\theta_j', a_j')$ that are both in the support of $\sigma^*$ but have the following payoff difference

$$
\mathcal{U}_j^{rev}(\theta_j, a_j, a_{-j}^*, \mathcal{D}) - \mathcal{U}_j^{rev}(\theta_j', a_j', a_{-j}^*, \mathcal{D}) > \epsilon + \epsilon'
$$

for some $\epsilon' > 0$.

Now pick $T$ such that for all $t \geq T$ we have

$$|\bar{\sigma}_j^t - \sigma_j^*| \max_{\theta_j, a_j, a_{-j}} \mathcal{U}_j^{rev}(\theta_j, a_j, a_{-j}, \mathcal{D}) \leq \frac{\epsilon'}{2K}$$

where $K$ is the number of pure strategy profiles. Such a $T$ exists since by assumption $(\bar{\theta}, \bar{a})$ converges and $\mathcal{U}_j^{rev}$ is bounded. We then have

$$
\begin{aligned}
\mathbb{E}[\mathcal{U}_j^{rev}(\theta_j', a_j', \bar{a}_{-j}^t, \mathcal{D})] &= \sum_{(\theta_{-j}, a_{-j})} \mathcal{U}_j^{rev}(\theta_j', a_j', a_{-j}, \mathcal{D}) \bar{\sigma}^t(\theta_{-j}, a_{-j}) \\
&\leq \sum_{(\theta_{-j}, a_{-j})} \left[ \mathcal{U}_j^{rev}(\theta_j', a_j', a_{-j}, \mathcal{D}) \bar{\sigma}^t(\theta_{-j}, a_{-j}) + \frac{\epsilon'}{2K} \right] \\
&\leq \sum_{(\theta_{-j}, a_{-j})} \mathcal{U}_j^{rev}(\theta_j', a_j', a_{-j}, \mathcal{D}) \bar{\sigma}^t(\theta_{-j}, a_{-j}) + \frac{\epsilon'}{2} \\
&< \sum_{(\theta_{-j}, a_{-j})} \mathcal{U}_j^{rev}(\theta_j, a_j, a_{-j}, \mathcal{D}) \bar{\sigma}^t(\theta_{-j}, a_{-j}) - \frac{\epsilon'}{2} - \epsilon \\
&\leq \sum_{(\theta_{-j}, a_{-j})} \left[ \mathcal{U}_j^{rev}(\theta_j, a_j, a_{-j}, \mathcal{D}) \bar{\sigma}^t(\theta_{-j}, a_{-j}) + \frac{\epsilon'}{2K} \right] - \frac{\epsilon'}{2} - \epsilon \\
&\leq \sum_{(\theta_{-j}, a_{-j})} \mathcal{U}_j^{rev}(\theta_j, a_j, a_{-j}, \mathcal{D}) \bar{\sigma}^t(\theta_{-j}, a_{-j}) - \epsilon \\
&= \mathbb{E}[\mathcal{U}_j^{rev}(\theta_j, a_j, \bar{a}_{-j}^t, \mathcal{D})] - \epsilon
\end{aligned}
$$

Thus we have that after iteration $T$ we no longer select $(\theta_j', a_j')$ since it is not within the set of $\epsilon$ best responses. This follow from the above algebra and the fact that $\mathcal{U}_j^{rev}$ is bounded above by zero (since it is the negative maximum regret).

But this implies that thus

$$\lim_{t \to \infty} \bar{\sigma}^t(\theta_{-j}, a_{-j}) \to 0.$$

But this is a contradiction since we assumed $\sigma^*(\theta_j', a_j') > 0$.

Now we prove local optimality.

Suppose we do not have local optimality. Then there exists $j$ and revelation game actions $(\theta_j, a_j)$ such that

$$\mathbb{E}[\mathcal{U}_j^{rev}(\theta_j, a_j, a_{-j}^*, \mathcal{D})] + \epsilon' < \epsilon$$

and

$$V(\theta_j, a_j, \sigma_{-j}^*) + \epsilon' < V(\sigma^*)$$

for some $\epsilon' > 0$.

Since the expected value of $V$ is continuous in the empirical distribution there exists $(\theta_j', a_j')$ with $\sigma^*(\theta_j', a_j') > 0$ such that $V(\theta_j, \bar{\theta}_{-j}, a_j, \bar{a}_{-j}^t) + \epsilon'' < V(\theta_j', \bar{\theta}_{-j}, a_j', \bar{a}_{-j}^t)$ for all $t \geq T'$ for some sufficiently large $T'$.

Now pick $T \geq T'$ such that for all $t \geq T$, $(\theta_j, a_j)$ is in the $\epsilon$-best-response set to $\sigma_{-j}^*$ for $j$. Such a $T$ is guaranteed to exist by continuity of $V$ and $\mathcal{U}_j^{rev}$ in the empirical distribution. But then best responses never select $(\theta_j', a_j')$ after $T$ which is a contradiction. $\qquad \square$

## 5 Additional Analysis for Auction Experiments

The top panel of Figure 1 plots the RMAC estimated types as a function of true type and we can see that the type distribution is fairly uniformly shifted down. As a robustness check we can also see that this downward shift is not affected by the counterfactual game. This is not a general property

Figure 1: In depth analysis of how RMAC changes counterfactual estimates. Top panel shows estimated types and RMAC pushes the entire distribution up or down, in this special case the extent of the downward shift is not affected by the counterfactual game. This happens in auctions because in RMAC the type regret is determined by $\mathcal{G}$ and $\mathcal{D}$ and lower valuations will guarantee lower $V$ in the counterfactual game. Bottom panel shows RMAC generated counterfactual strategies for various counterfactual auctions.

of the RMAC estimator, and is specific to this case of auctions where revenue will be monotonic in counterfactual bid and counterfactual bid will be monotonic in type.

The worst case scenario is compounded by assumption that the equilibrium that attains in the counterfactual will be the one where these same individuals will slightly underbid. We can see the RMAC type-contingent counterfactual strategies plotted in the bottom panel of figure 1. Error ribbons reflect $10^{th}$ and $90^{th}$ percentiles taken over multiple replicates with wide bands appearing when reserves are set high since any bid below the reserve always achieves a payoff of $0$ and so individuals are indifferent between those bids.

## 5.1 RMAC Without Point Identification

We now discuss how RMAC can be useful for situations where point identification of a structural model is not guaranteed. This can happen when there are multiple equilibria in $\mathcal{G}'$ or when the mapping from type distributions to equilibrium distributions in $\mathcal{G}$ is not injective. In such situations there will be multiple solutions to a maximum likelihood estimator and no guarantees about which one will be output by the procedure. On the other hand, RMAC bounds will still be well defined and if we choose a small enough $\epsilon$ will be close to the worst and best case full equilibria.

We illustrate this by considering counterfactual prediction where $\mathcal{G}$ is a 2 player second-price auction with reserve .5. with the same simulation parameters as above (as $\mathcal{D}$ we use truthful reports). $\mathcal{G}$ is dominant strategy truthful for all types $\theta > .5$ but the payoff to bids in the interval $[0, .5]$ is always 0 so any type $\theta < .5$ can rationalize any bid in this interval. This means that the type distribution is not point identified from an action distribution. We apply RMAC to this situation with the counterfactual question of what would happen if we changed the reserve $r$.

Figure 2 shows the results. We see on the left panel that RMAC bounds for reserves $[0, .5]$ are very wide whereas bounds for reserves above the original .5 are smaller since our type censoring appears only on one side.

The right panel shows that here, unlike in the auction experiments above, the choice of $\mathcal{G}'$ does affect type estimation. When the counterfactual reserve is 0 then the pessimistic RMAC pushes previously unidentified types to 0 to create the worst case $\mathcal{G}'$ equilibrium. When the counterfactual reserve is set very high to .9 low types do not bid above the reserve even in the optimistic $\epsilon$ equilibria and so types which were not identified in the original $r = .5$ game remain unidentified and their guesses are chosen arbitrarily.

Figure 2: Results for data drawn from a second-price auction with reserve .5 with counterfactual question involving changing the reserve. RMAC is well defined even when the inverse problem is not identified due to multiple types being consistent with the same observed actions. The maximum likelihood solution (red line) simply picks a random type from among all equally likely ones. RMAC bounds reflect the lack of identification in the original game as they are quite large for counterfactual reserves less than the original reserve. In the right panel we see that in this situation, unlike in the example above, the choice of counterfactual game $\mathcal{G}'$ does affect the estimated underlying types.

## 6 RMAC in Social Choice

As our last study we move to the domain of social choice. We consider the standard example of a group of individuals choosing an ideal point $x^* \in [0, 1]$. We assume individuals have a type $\theta \in [0, 1]$ and have single-peaked preferences and receive loss $(x^* - \theta)^2$ from a point $x^*$ being chosen for the group. We consider groups of 11 individuals participating in one of 3 mechanisms: in each mechanism individuals report a number $a \in [0, 1]$. In the *mean* mechanism, $x^*$ is chosen as the mean of the reports, in the *median* mechanism the median is chosen. In both the median and mean mechanism no side payments are made.

Figure 3: The median and VCG mechanisms are dominant strategy truthful but they have very different robustness properties. The mean mechanism is not well identified as types outside a narrow interval all report extreme values in equilibrium, however, RMAC bounds are defined for this case as well.

In the *VCG* mechanism individuals pay the mechanism their externality on everyone else (i.e. the difference in total utility from choosing the mean that includes the report of $a_i$ and the one that excludes it) and the mean is chosen. As in auctions we discretize the types and actions with a grid of .01. We sample 1000 types, calculate their optimal action in the mechanism, and run the revelation game. For the counterfactual valuation we use $V(\hat{\theta}) = \sum_i \theta$. That is, we look for the most right or left shifted type distributions that are consistent with observed data.

Figure 3 shows our results. First, we can see that with $\epsilon = 0$ the mean mechanism is not identified since in equilibrium a whole range types choose the 0 and 1 actions. However, even with small $|\epsilon|$ the solution becomes unique.

Even though both the median and VCG mechanisms are dominant strategy truthful they have very different robustness properties. In the median mechanism deviation from truthful reporting, in particular for extreme types, is not very costly as it can only affect outcomes if that person is pivotal. On the other hand, in the VCG any deviation also changes the price one has to pay into the mechanism, thus changing the way types can deviate under RMAC.

## References

Vincent Conitzer and Tuomas Sandholm. 2008. New complexity results about Nash equilibria. *Games and Economic Behavior* 63, 2 (2008), 621–641.