[Reviews · NeurIPS 2019]

Reviewer 1



# Summary The paper presents a framework for calculating bounds on the value of counterfactual multi-agent games using only logged data of actions that occurred in a factual game. This problem arises in a number of mechanism design contexts, where intervening on a system constitutes changing the rules of the game. Calculating counterfactual value requires reasoning about how rule changes affect equilibrium behavior of the agents. Under strong assumptions this counterfactual value is point-identified, but these assumptions are often implausible. The authors present a scheme for relaxing these assumptions, and characterizing the set of values that are compatible with the observed data under this relaxation. The relaxation of point-identification assumptions is presented in terms of a second game, which the authors call the Revelation Game. When the standard identifying assumptions hold, the equilibrium of this game recovers agent types as well as counterfactual actions. The authors relax the standard identifying assumptions by relaxing the equilibrium conditions of this game by some constant \epsilon, which corresponds to relaxing the equilibrium conditions of the observed game and/or the counterfactual game by \epsilon. The authors then discuss methods for seeing out the extreme values of the value function attained among the (type, counterfaction action) pairs within this \epsilon-equilibrium set. In general, this problem is NP-hard, but the authors show that the Fictitious Play heuristic can be extended to the revelation game to obtain locally extreme values in cases where the fictitious play iteration converges. Finally, the authors demonstrate computation of the RMAC bounds in some simple examples. # Feedback Overall, this paper was a pleasure to read. The paper is very clearly written, and as somebody from an adjacent focus (causal inference), the paper did a great job of explaining the necessary game-theoretic concepts to get me up to speed on its own. The problem is well-motivated, and the approach is compelling, although I think the scope should be narrowed a bit. The experiments/demos could have been a bit more compelling, but I chalk some of this up to space constraints. Comments on broader points that could improve the paper follow. ## The Revelation Game The one part of the paper where I thought the exposition fell a little short was the description of the Revelation Game. After my second read through the paper, I think I was able to get what was going on, but the first time through it was a little bit difficult to parse. I think the presentation of the game could be better motivated. For example, later on line 164, you seem to indicate that relaxations of the game correspond to relaxations of the factual and counterfactual games, respectively. Could you use this to motivate the revelation game loss function directly? I think it would also make the interpretation of \epsilon clearer from the get-go. As the paper is now written, I left this section with two unanswered questions: What do each of the regret expressions represent? Why are they aggregated using a max, rather than, say, a weighted average? I think these questions are related to a larger question that I have, which is, why is relaxing the revelation game the right relaxation? I personally feel like it’s reasonable, but it seems like there are many ways to parameterize sets of (types, counterfactual actions) that shrink to a point when the standard assumptions hold. ## Which assumptions are being relaxed? The authors indicate (e.g., line 166) that relaxing the revelation game corresponds to relaxing assumptions 1-4, but this statement is vague. While technically true that the conjunction of these assumptions is being relaxed, it seems that \epsilon is only relevant for the equilibrium Assumption 1 and (potentially) the specification Assumption 4. For example, if Assumption 2 were violated, one could still characterize the set of (types, counterfactual actions) that are equally supported by the observed data without having any notion of an \epsilon-BNE for the revelation game (this is essentially the argument that is made in the last paragraph of the school choice example). Likewise, if Assumption 3 were violated, one could enumerate the set of equilibria in the counterfactual game without reference to the revelation game. I personally think that it would be wise to claim that the epsilon-RMAC method is really only relaxing Assumptions 1 and 4. Actually, I think it would probably make sense to remove Assumption 4 because I’m not sure that the \epsilon-BNE would even contain the equilibria implied by the correct reward functions (misspecification can be really bad). On the other hand, it’s clear how the \epsilon relaxation relates to the equilibrium assumption. As far as the identification assumptions go, I think that it’s fine to say that the RMAC bounds are _compatible_ with violations of Assumptions 2 and 3. This is closer to the statement that’s made in the contributions section of the introduction. ## Interpretation of \epsilon It would be useful to have more explicit discussion of the interpretation of \epsilon. Can you give concrete examples of how \epsilon related directly to relaxations of the standard assumptions? There are some hints along this vein (paragraph starting on line 164, the mention of “misoptimization/misspecification” on line 252), but no formal discussion. In particular, how do we know what reasonable values of \epsilon are? Are there empirical quantities that we can anchor on? Is it possible to separately reason about each source of fragility (misoptimization vs misspecification)? Again, the auctions example touches on this, but to me interpreting epsilon requires a much deeper discussion. I ask these questions drawing parallels to the literature on sensitivity analysis in causal inference that explores deviations from the unconfoundedness/ignorability/backdoor criterion assumption in observational studies. Here, one of the standard tasks is to tie the sensitivity parameters (which play the same role as epsilon here) to observable quantities such as variance in treatment assignment or outcome explained by observed variables. No need to cite these, but some examples of this type of argument appear in: Cinelli and Hazlett: https://polmeth.byu.edu/Plugins/FileManager/Files/Papers/making_sense_of_sensitivity_10July2018.pdf Franks et al: https://arxiv.org/abs/1809.00399 Imbens: https://www.aeaweb.org/articles?id=10.1257/000282803321946921 To this end, it might be useful to see an example where you actually instantiate a population of faulty agents, to show how their “faults” in optimization or specification translate to an \epsilon, and to show that the RMAC bounds include the value associated with the game played with that population. This would help clarify the mapping from \epsilon to concrete relaxations of the standard assumptions, and would be closer to an “end to end” test of the method in something resembling a real application. ## Computing Equilibria Could you be more explicit about what the gap is between the NP-hard exact solution to the problem and the RFP solution? I am gathering that there are cases where the RFP may not converge, and where the locally V-optimal solution may not be globally V-optimal. It might be useful to pull these together so the reader can understand quickly what is lost by moving to the first-order RFP solution. ## Experiments I like the auctions experiment a lot. I think I would be more interested in seeing this example treated in more depth, rather than seeing the school choice experiment (which would be fine in the appendix). Per my discussion above about which assumptions are being relaxed, I think the auctions experiment actually explores the implications of having nonzero epsilon, whereas the school choice experiment is largely concerned with assumptions that don’t require the revelation game / epsilon-BNE formalism at all (you’re only relaxing the identification assumption, so you could get all of your results with epsilon = 0). To this end, I think it would make more sense to give a more thorough treatment of the auctions example (e.g., by adding the concrete “faulty agent” simulation I suggested above), and to show that RMAC is compatible with violations of Assumption 2 by moving the example from the appendix into the main text. ---------------------------- # Post-Rebuttal and Discussion Edit: Following discussion with the other reviewers, I'd like to reiterate that I think the exposition of the revelation game really does need to be clarified. The other reviewers convinced me that the writing and notation in that section was a bit too convoluted and vague, especially for the section that presents the key contribution of the paper. They convinced me to temper my recommendation a bit, but I still think this paper is an Accept. I also stand by my assessment that the school choice example probably doesn't belong in the main text because the lack of identification there doesn't really fit into the epsilon-BNE framework. So I do think that it would still make sense to have an example where you show that a population of sub-optimal agents would in fact induce values contained in the RMAC bounds.

Reviewer 2



Update: Based on the authors' response, I am not convinced that my main concerns will be addressed adequately. I will, therefore, maintain my current recommendation. Originality: As far as I know the approach proposed is innovative –– it uses ideas from the partial identification literature, but adapts them to a new setting in a novel way. Quality / Clarity: the paper has a number of issues concerning rigor and clarity. The authors try to explain concepts in words instead of drowning the reader in mathematical formalism –– this is generally a good thing. Unfortunately, this paper falls into the other excess: some important objects are either undefined or defined ambiguously. - What is the mathematical structure of $\mathcal{D}$? What does an element of $\mathcal{D}$ look like? Does it contain information on a single game, or multiple games? Are the types observed and part of $\mathcal{D}$? - On line 132 you have an object $d_i$ that, as far as I can see, is undefined. You call it an action but up to that point actions where denoted $a_i$. You use $d_i$ again when writing down $Regret^{\mathcal{G}}_j$, as if it were an action. - As defined in Definition 1, a utility function takes $N+1$ arguments: N actions and a type (the type coming last). When you use utilities to define the regret, however, the first argument is an action, the second is a type and the third argument, \mathcal{D}_j$ is... well, not defined formally anywhere. Please clarify. - You sort of define it in the text but I that the concept of $\epsilon$-BNE deserved a proper definition environment. - I also found the revelation game somewhat unhelpful for the exposition. I is not clear to me how it connects to problem at hand. Significance: I think that the objective of the paper – relaxing excessively strong assumptions usually made in the literature – is an important one, and any contribution in that direction is welcome.

Reviewer 3



UPDATE: I thank the authors for their comments. I decided to update the score upwards as I expect that all issues pointed out by the reviewers can be addressed in revision. This papers deals with the counterfactual prediction problem in settings with multi agent systems. The problem is that if we change the rules of the game, then the system may try to adapt to the new rules. Estimating the effects of this adaptation requires taking into account the incentive structure of the game. Overall, I find the paper interesting and the applications useful. The paper is well written and the main ideas are clearly presented. At the same time, I have some concerns: 1) There is no adequate comparison with existing ideas and methods in the literature. For example, it is not explained how this method differs practically from [1] and [2]. In [1] the authors discuss the inference problem and propose methods to quantify the uncertainty of the policy change, similar to the confidence bands derived by the authors in this paper. In [2] the setting is also very similar to the setting in the current paper: how to estimate the long-term effects of the policy change on outcomes (e.g., auction revenue). The general approach in [2] also looks very similar to what is proposed in Section 4 (and summarized in Fig 4). Both papers also give consistency results under some assumptions. I understand that the goal in this paper is different, namely, to address robustness to rationality assumptions. However, the end result in all papers is similar, that is, confidence bands on the effects of the policy change. It would help then to understand what are the comparative benefits and limitations of each method. 2) Perhaps with the exception of Theorem 3, there appears not to be much technical contribution in the paper. Not necessarily a crucial problem, but it would be nice to know what new techniques/methods may be more broadly interesting. Minor comment: The sentence is broken in Page 4. [1] "Counterfactual reasoning and prediction in learning systems: The example of computational advertising", Bottou et al. [2] "Long-term causal effects via behavioral game theory", Toulis, Parkes

[Author Response · NeurIPS 2019]

**Reviewer 1** *"What do...regret expressions represent? Why...max?"* The regret expressions represent how suboptimal agents are in each of the games (i.e. how much assumptions 1,4 in each of the games are violated). They are aggregated with a max since we would like to allow an $\epsilon$-BNE in both $\mathcal{G}$ and $\mathcal{G}'$. Aggregating, for example, with a sum would mean that we allow for $\epsilon_1$ and $\epsilon_2$ equilibria in the games where $\epsilon_1 + \epsilon_2 \leq \epsilon$. We will make this clearer in the text.

*"why is...revelation game the right relaxation?"* We agree many relaxations are possible. We think the revelation game is attractive for the use-case of mechanism design since here the analyst knows the game structure clearly (they have coded the mechanism) but only has a model of the agents (possible utility functions, how well the agents optimize) and wants to be robust to the agent model being wrong.

*"really only relaxing Assumptions 1 and 4... not sure that the $\epsilon$-BNE...contain the equilibria implied by the correct reward functions"* We agree with the reviewer's discussion of the relationship between RMAC and assumptions 1,2,3. We will clarify the text. Re: assumption 4, we agree that the fact that our procedure can be thought of as misspecification is not necessarily obvious, however, we have proven the following result: *Let $(\mathcal{G}, \mathcal{G}')$ be the real game/counterfactual game, let $(\mathcal{G}_m, \mathcal{G}'_m)$ be misspecified versions of these two games with same type/action spaces but $||u_m - u||_\infty \leq \frac{\epsilon}{2}$ (and same for $u'$). Let $\mathcal{D}$ be some data. If $r^* = (\hat{a}, \hat{\theta})$ is an equilibrium of the real revelation game corresponding to $(\mathcal{G}, \mathcal{G}', \mathcal{D})$ then $r^*$ is also an $\epsilon$-equilibrium of the misspecified revelation game corresponding to $(\mathcal{G}_m, \mathcal{G}'_m, \mathcal{D})$.* Note the converse is not true, so RMAC is a pessimistic estimate of relaxing assumption 4 appropriate for real world cases where we care about e.g. worst-case revenue. We will add this as a formal theorem to the paper.

*"It would be useful to have more explicit discussion of the interpretation of $\epsilon$.* $\epsilon$ actually often has a natural scale since in many applications the units of the utility function have a natural scale. For example, in the case of auction where valuations are in dollars an $\epsilon = .5$ means individuals can behave in a suboptimal way that loses up to 50 cents relative to their optimum. We will add this example into the text.

*"useful to see an example...[with] population of faulty agents..."* The school choice example can actually be thought as doing this. The actions that we observe in the population can come from either a set of strategic agents or naive truthful agents. Depending on which underlying population generated the original data, we will get different counterfactual conclusions. The RMAC bounds show precisely this. We will expand the discussion to make this point explicitly.

*"gap... between NP-hard exact solution..and RFP"* Our results guarantee that if RFP converges then it converges to a local optimum. In some cases, this may not be the global optimum. Unfortunately, the MIP to compute the global optimum has $|\mathcal{D}|^2$ boolean variables so computing the exact solution is infeasible for instances beyond 10-20 data points. For such small cases a gap (or lack of gap) is unlikely to be representative of real world problems. One useful datapoint from our experiments is that across multiple initializations we found identical results in our auction/social choice experiments so it seems as though there are not multiple minima in these domains.

**Reviewer 2** *"What does an element of $\mathcal{D}$ look like...single game, or multiple games?..types observed?'* $d_i$ is an action played by an agent in the a single instance of the game. For example, if we consider the 'what would happen if we raised the reserve in the auction?' scenario then $\mathcal{D}$ each consists of an agent's bid when from one auction. In this paper we deal with 1 action per agent (though this is not required for the algorithm/theory). Importantly, types are **never** observed and must be inferred from data by making assumptions (in a standard model these are assumptions 1,2,4). We apologize this was unclear and will clarify the text.

*"utilities to define the regret [not well defined]."* We apologize and will fix this and other raised missing formalism. $d_j$ is the action taken by the individual in $\mathcal{G}$. $\mathcal{D}_{-j}$ is the distribution of actions observed from everyone else. The regret for $\mathcal{G}$ of an individual with estimated type $\hat{\theta}_j$ is then the amount they lose by taking $d_j$ instead of type $\hat{\theta}_j$'s optimal action (given that everyone else is behaving according to $\mathcal{D}_{-j}$). The regret for the counterfactual game is analagous except using the estimated counterfactual actions. We will make this clearer in the text.

**Reviewer 3** *"not explained how this method differs practically from [1] and [2]"* We cite [1] in the text already and will add a citation to [2]. Both of these papers make their own versions of assumptions 1-4 - in other words, they can both be thought of as the solid arrow in Figure 4. By contrast, our goal is to relax these strong assumptions. As we discuss in the related work approaches like [1],[2] "allow for measures of statistical uncertainty..[but do] not allow analysts to check for robustness of conclusions to violations of assumptions."

We **do** make the comparison to the standard approach: we show the pure statistical uncertainty estimates (example: gray ribbons around $\epsilon=0$ line in Figure 1). One can see that the statistical uncertainty in the counterfactual estimate is quite small relative to the non-robustness to small violations of the assumptions. We will make this clearer in the text.

*"...it would be nice to know what new techniques/methods may be more broadly interesting."* We provide 1) a MIP for small games, 2) a first-order method with some guarantees. Both of these are more broadly applicable than just our experiments and lead to many open theoretical questions.

[Meta-Review · NeurIPS 2019]

Reviewers found this paper to be an original and useful addition to the field of multi-agent games. While some of the presentation could be clarified (see specific reviewer comments, e.g. about the revelation game), there was a consensus that the paper is generally well-written and clear enough for publication, with the proposed corrections.